

# Ecological and molecular characterization of a coral black band disease outbreak in the Red Sea during a bleaching event

Ghaida Hadaidi[1], Maren Ziegler[1], Amanda Shore-Maggio[2], Thor Jensen[1], Greta Aeby[3] and Christian R. Voolstra[1]

[1] Red Sea Research Center, Division of Biological and Environmental Science and Engineering (BESE), King Abdullah University of Science and Technology (KAUST), Thuwal, Saudi Arabia
[2] Institute of Marine and Environmental Technology (IMET), University of Maryland, Baltimore County, Baltimore, MD, United States of America
[3] Hawai'i Institute of Marine Biology, Kāne'ohe, HI, United States of America

Corresponding authors
Greta Aeby, greta@hawaii.edu
Christian R. Voolstra,
christian.voolstra@kaust.edu.sa

## ABSTRACT

Black Band Disease (BBD) is a widely distributed and destructive coral disease that has been studied on a global scale, but baseline data on coral diseases is missing from many areas of the Arabian Seas. Here we report on the broad distribution and prevalence of BBD in the Red Sea in addition to documenting a bleaching-associated outbreak of BBD with subsequent microbial community characterization of BBD microbial mats at this reef site in the southern central Red Sea. Coral colonies with BBD were found at roughly a third of our 22 survey sites with an overall prevalence of 0.04%. Nine coral genera were infected including *Astreopora*, *Coelastrea*, *Dipsastraea*, *Gardineroseris*, *Goniopora*, *Montipora*, *Pavona*, *Platygyra*, and *Psammocora*. For a southern central Red Sea outbreak site, overall prevalence was 40 times higher than baseline (1.7%). Differential susceptibility to BBD was apparent among coral genera with *Dipsastraea* (prevalence 6.1%), having more diseased colonies than was expected based on its abundance within transects. Analysis of the microbial community associated with the BBD mat showed that it is dominated by a consortium of cyanobacteria and heterotrophic bacteria. We detected the three main indicators for BBD (filamentous cyanobacteria, sulfate-reducing bacteria (SRB), and sulfide-oxidizing bacteria (SOB)), with high similarity to BBD-associated microbes found worldwide. More specifically, the microbial consortium of BBD-diseased coral colonies in the Red Sea consisted of *Oscillatoria* sp. (cyanobacteria), *Desulfovibrio* sp. (SRB), and *Arcobacter* sp. (SOB). Given the similarity of associated bacteria worldwide, our data suggest that BBD represents a global coral disease with predictable etiology. Furthermore, we provide a baseline assessment of BBD disease prevalence in the Red Sea, a still understudied region.

## INTRODUCTION

The rise of coral disease outbreaks contributes to the decline of coral reefs globally (*Cróquer & Weil, 2009*; *Harvell et al., 2009*; *Hoegh-Guldberg, 2012*; *McLeod et al., 2010*; *Randall &*

*Van Woesik, 2015*) and coral disease appears to be the most destructive factor on many reefs. For instances, the Caribbean has been named a "disease hot spot" due to the fast emergence, high prevalence, and virulence of coral diseases in this region (*Rosenberg & Loya, 2013*). Coral disease outbreaks in the last decades in the Caribbean have resulted in significant losses in coral cover, diversity, and habitat (*Aronson & Precht, 2001*; *Bruckner, 2002*; *Hughes, 1994*; *Precht et al., 2016*; *Weil, 2002*). Following the mass-bleaching event in 2005 in the US Virgin islands, coral disease outbreaks reduced coral cover by more than 50% (*Cróquer & Weil, 2009*; *Miller et al., 2009*).

Coral diseases were first reported in the Caribbean in the 1970s, including black band disease (BBD), which is considered the most studied coral disease due to its widespread occurrence on reefs around the world (*Bourne, Muirhead & Sato, 2011*; *Richardson, 2004*). Black band disease has been reported from reefs throughout the Caribbean, the Indo-Pacific regions, the Red Sea, and the Great Barrier Reef (*Al-Moghrabi, 2001*; *Dinsdale, 2002*; *Green & Bruckner, 2000*; *Kaczmarsky, 2006*; *Lewis et al., 2017*; *Montano et al., 2012*; *Page & Willis, 2006*; *Sutherland, Porter & Torres, 2004*; *Weil et al., 2012*). BBD is the first described coral disease (*Antonius, 1973*), affecting scleractinian and gorgonian corals (*Green & Bruckner, 2000*; *Sutherland, Porter & Torres, 2004*; *Weil, 2004*). BBD prevalence generally is considered low (*Dinsdale, 2002*; *Edmunds, 1991*; *Weil, 2002*); however, this disease is a serious threat to coral reef ecosystems worldwide due to its persistence, leading to coral mortality in the long-term (*Bruckner & Bruckner, 1997*; *Green & Bruckner, 2000*; *Kaczmarsky, 2006*; *Kuta & Richardson, 1996*; *Page & Willis, 2006*; *Sutherland, Porter & Torres, 2004*; *Zvuloni et al., 2009*). Susceptibility to BBD differs between coral taxa and may result in long-term changes to coral community structure (*Bruckner & Bruckner, 1997*; *Page & Willis, 2006*). The abundance of BBD is affected by several environmental factors, including seawater temperature, water depth, solar irradiance, host population diversity, and anthropogenic nutrients (*Al-Moghrabi, 2001*; *Kaczmarsky, 2006*; *Kuta & Richardson, 2002*; *Montano et al., 2013*). Interestingly, seasonal temperatures influence BBD prevalence, with increased virulence during warmer summer months (*Richardson & Kuta, 2003*; *Rützler & Santavy, 1983*; *Willis, Page & Dinsdale, 2004*), as for example in the Maldives where sea surface temperatures above 28 °C promoted BBD infections (*Montano et al., 2013*).

BBD manifests as a dark band that migrates across the coral colony at a rate of >1 cm/day (*Richardson, 1998*) leaving behind bare skeleton. The base of the BBD mat is anoxic and high in sulfide levels, causing damage and necrosis to coral tissue (*Ainsworth et al., 2007*; *Carlton & Richardson, 1995*; *Richardson et al., 1997*). The BBD mat is composed of a polymicrobial consortium, dominated by filamentous cyanobacteria, sulfate-reducing bacteria (SRB), including members of *Desulfovibrio* spp., sulfide-oxidizing bacteria (SOB) (*Beggiatoa* spp.), and other heterotrophic bacteria (*Cooney et al., 2002*; *Miller & Richardson, 2011*; *Sato, Willis & Bourne, 2010*). As a result of diel light changes, the microbial members of the BBD mat undergo vertical migrations, which causes the harmful microenvironment on top of the coral tissue (*Carlton & Richardson, 1995*; *Miller & Richardson, 2011*; *Richardson, 1996*). Oxygen depletion and high sulfide concentrations are produced by SRB, which is lethal to the coral tissues and considered the most important factor in BBD pathogenicity (*Glas et al., 2012*; *Richardson, 1996*; *Richardson et al., 1997*; *Richardson et al., 2009*). Although

the functional composition of the BBD mat is conserved, the diversity of the microbial consortium in BBD differs according to geographic location and coral species (*Cooney et al., 2002*; *Frias-Lopez et al., 2004*; *Sekar et al., 2006*).

The occurrence of BBD in the Red Sea was first recorded by *Antonius (1988)* where the severity of BBD was measured from rare to moderate and mostly correllated with elevated temperatures and seawater pollution. However, baseline data on BBD prevalence in the Red Sea is still lacking. To fill this gap, we conducted surveys to determine the distribution and prevalence of BBD across central Red Sea reefs spanning 4 degrees of latitude. We also detected a bleaching-associated outbreak of BBD on a coral reef in the southern central Red Sea and characterized the microbial community of BBD microbial mats from *Coelastrea* sp., *Dipsastraea* sp., *Goniastrea* sp., and *Platygra* sp. using high-throughput sequencing. We compared the microbial consortium to that reported from other regions of the world in order to identify biogeographic patterns in the main BBD consortium members.

## MATERIAL AND METHODS

### Black band disease surveys

Coral community structure and BBD prevalence was recorded at 22 sites spanning approx. 535 km along the coast of Saudi Arabia in the Red Sea (Fig. 1, Table 1). At least six reefs per region (Yanbu, Thuwal, Al-Lith) were surveyed with three additional reefs in Thuwal and one reef in Jeddah (80 km from Thuwal) that were surveyed as time permitted. The reefs sampled/assessed in this study do not fall under any legislative protection or special designation as a marine/environmental protected area. Under the auspices of KAUST (King Abdullah University of Science and Technology), the Saudi Coastguard Authority issued sailing permits to the sites that include coral collection. At each site, divers counted coral colonies by genera along two replicate belt transects (25 m × 1 m). At the same time, point-intercept method was used to characterize the substrate at 25 cm intervals. All corals with BBD lesions were identified along wider 25 × 6 m transects and photographed. Depending on depth and time limits, the length of transects were adjusted as necessary. Survey sites ranged in depth between 3 and 7.6 m and all surveys were conducted from 19 October to 3 November 2015. The diver surveys were used to determine average percent coral cover, coral community composition, and colony densities. Underwater time constraints prevented counting all colonies within the larger 25 × 6 m belts surveyed for disease. Therefore, BBD prevalence was estimated by calculating the average colony density (by genus) within the 25 × 1 m transect and then extrapolating the colony counts to the wider 25 × 6 m disease survey area and using this as the denominator of prevalence calculations, i.e., (number of colonies with BBD lesions/total number of estimated colonies) * 100) (*Aeby et al., 2015a*). At the outbreak site, diseased coral colonies were so numerous that only 49 m$^2$ of the transect could be surveyed. The frequency of disease occurrence (FOC) was calculated by dividing the number of sites having corals with BBD lesions by the total number of sites surveyed. At the localized BBD outbreak site a chi-square goodness-of-fit test was used to examine differential distribution of the number of BBD versus healthy colonies among the coral genera affected by the disease. The chi-square test
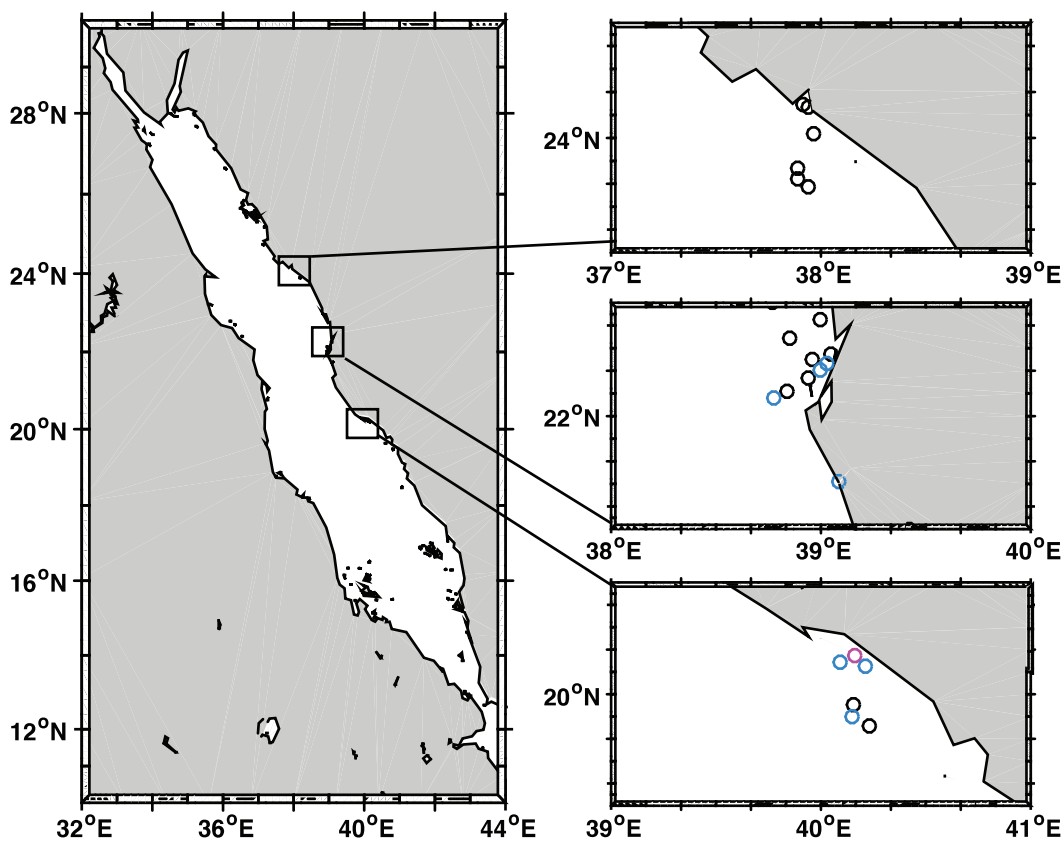

**Figure 1** **Black band disease survey locations of 22 reef sites along the central Red Sea coast of Saudi Arabia.** Survey points marked for Yanbu region (north), Thuwal (central), and Al-Lith (south). Sites without black band diseased coral colonies marked in black, sites with one or two diseased colonies in blue, and the site where a localized outbreak of BBD was observed is marked in pink.

compares the observed vs. expected number of infected colonies based on the abundance of each coral genus in the field. Statistical analysis was performed using JMP statistical software (v. 10.0.2, SAS Institute Inc., Buckinghamshire, UK).

## Sample collection of black band disease microbial mats and 16S rRNA gene sequencing

Microbial mats were collected from BBD infected coral genera (one colony of *Coelastrea* sp., two colonies of *Dipsastraea* sp., three colonies of *Goniastrea* sp., and one colony of *Platygra* sp.) at the site of the observed BBD outbreak (Al-Lith fringing reef 1) in November 2015. Microbial mats were siphoned off the coral surface with Pasteur pipettes and transferred into ziplock bags under water. Sample replication was limited by obtainable coral species on this reef site due to environmental conditions.

Samples were homogenized using bead-beating via TissuLyser II (Qiagen, Hilden, Germany) twice for 30 sec at 30 Hz, 20 μl of the homogenate were boiled in sterile Milli-Q water at 99 °C for 5 min and subsequently 1 μl was directly used as PCR template. To amplify the variable region 4 of the 16S rRNA gene, the following primers were

Hadaidi et al. (2018), *PeerJ*, DOI 10.7717/peerj.5169

**Table 1** Survey of black band disease (BBD)-affected coral colonies at 22 reef sites in the central Red Sea. Coral genus counts denote number of BBD-affected colonies.

| Region | Reef site | GPS (latitude, longitude) | Depth (m) | Area colony count survey (m$^2$) | Area BBD survey (m$^2$) | Montipora | Dipsastraea | Psammocora | Gardineroseris | Astreopora | Pavona | Platygyra | Coelastrea[a] | Goniopora[a] | Total no. of BBD | Total no. surveyed | BBD prevalence (%) |
|---|---|---|---|---|---|---|---|---|---|---|---|---|---|---|---|---|---|
| Yanbu | Marker 32 | 23.8664, 37.8913 | 4 | 25 | 75.6 | | | | | | | | | | 0 | 726 | 0 |
| | Marker 35 | 23.8207, 37.9350 | 4.6 | 25 | 78 | | | | | | | | | | 0 | 930 | 0 |
| | Abu Galaba | 23.7891, 37.9393 | 3 | 21 | 61.8 | | | | | | | | | | 0 | 848 | 0 |
| | Fringing reef 1 | 24.1362, 37.9396 | 5.2 | 25 | 150 | | | | | | | | | | 0 | 1,308 | 0 |
| | Marker 10 | 24.0189, 37.9666 | 4.6 | 25 | 126 | | | | | | | | | | 0 | 1,749 | 0 |
| | Fringing reef 2 | 24.1452, 37.9149 | 4.6 | 25 | 150 | | | | | | | | | | 0 | 1,830 | 0 |
| Thuwal | Abu Madafi | 22.0766, 38.7751 | 4 | 25 | 300 | | 1 | | | | | | | | 1 | 2,184 | 0.05 |
| | Al Fahal | 22.1119, 38.8411 | 4.6 | 23.5 | 300 | | | | | | | | | | 0 | 6,000 | 0 |
| | Al-Mashpah | 22.0772, 38.7744 | 6.7 | 25 | 300 | | | | | | | | | | 0 | 4,200 | 0 |
| | Inner Fsar | 22.2358, 39.0304 | 4.6 | 25 | 300 | | | | | 1 | | | | | 1 | 6,852 | 0.01 |
| | Shaab | 22.2012, 38.9992 | 4.6 | 25 | 300 | | | | 1 | | | | | | 1 | 5,778 | 0.02 |
| | Shi'b Nazar | 22.3409, 38.8521 | 4.9 | 23.5 | 300 | | | | | | | | | | 0 | 3,294 | 0 |
| | Tahlah | 22.2750, 39.0497 | 5.2 | 25 | 300 | | | | | | | | | | 0 | 3,780 | 0 |
| | Qita al Kirsh | 22.4257, 38.9957 | 4.6 | 25 | 300 | | | | | | | | | | 0 | 5,748 | 0 |
| | Um Alkthal | 22.1653, 38.9391 | 7.6 | 25 | 300 | | | | | | | | | | 0 | 5,208 | 0 |
| Jeddah | La Plage | 21.7092, 39.0832 | 4.6 | 25 | 300 | | | | | | | 1 | | | 1 | 474 | 0.21 |
| Al-Lith | Abu Lath | 19.9554, 40.1543 | 5.5 | 20 | 240 | | | | | | | | | | 0 | 5,556 | 0 |
| | South Reef | 19.8985, 40.1514 | 3.7 | 20 | 240 | | | | | 1 | | | | | 1 | 3,720 | 0.03 |
| | Al-Lith 3 | 19.8608, 40.2282 | 5.5 | 20 | 240 | | | | | | | | | | 0 | 4,320 | 0 |
| | Qita Al Kirsh | 20.1407, 40.0931 | 3 | 20 | 240 | | | | | 1 | | | | | 1 | 6,588 | 0.02 |
| | Fringing reef 1 | 20.1732, 40.1613 | 4.5 | 20 | 49 | 1 | 15 | 2 | | 1 | 1 | 1 | 1 | 1 | 23 | 1,281 | 1.72 |
| | Whaleshark reef | 20.1230, 40.2118 | 1.8 | 25 | 150 | | 1 | | 1 | | | | | | 2 | 1,716 | 0.12 |

**Notes.**

[a]Colonies of *Coelastrea* and *Goniopora* were only found outside the survey area.

used: 515F [5′TCGTCGGCAGCGTCAGATGTGTATAAGAGACAGGTGCCAGCMGCC GCGGTAA′3] and 806RB [5′ GTCTCGTGGGCTCGGAGATGTGTATAAGAGACAG GGACTACNVGGGTWTCTAAT′3] (*Apprill et al., 2015*; *Caporaso et al., 2012*; *Kozich et al., 2013*). Primer sequences contained sequencing adaptor overhangs (underlined above; Illumina, San Diego, CA, USA). Triplicate PCRs were performed for all samples with 0.2 μM of each primer in a total reaction volume of 25 μL using the Qiagen Multiplex PCR Kit. The following cycling conditions were used: 95 °C for 15 min, followed by 27 cycles of 94 °C for 45 s, 50 °C for 60 s, 72 °C for 90 s, and a final extension step of 72 °C for 10 min. Amplification was checked visually via 1% agarose gel electrophoresis. Triplicate samples were pooled and cleaned with ExoProStar 1-Step (GE Healthcare, Little Chalfont, UK). An indexing PCR was performed on the cleaned samples to add Nextera XT indexing and sequencing adaptors (Illumina, San Diego, CA, USA) following the manufacturer's protocol and followed by sample normalization and library pooling. 16S rRNA gene amplicon libraries were sequenced on the Illumina MiSeq platform using 2*300 bp overlapping paired-end reads with a 10% phiX control at the KAUST Bioscience Core Laboratory. Sequence data determined in this study are available under NCBI Bioproject ID: PRJNA436216.

## Sequence data processing and bacterial community analysis

Processing of raw sequence data was conducted in mothur (version 1.36.1; *Schloss et al., 2009*). Using the 'make.contigs' command, sequence reads were joined into contigs. Contigs longer than 310 bp and ambiguously called bases were excluded from the analysis. Subsequently, sequences that occurred only once across the entire dataset (singletons) were removed. The number of distinct sequences were identified and counted, and the total number of sequences per sample was determined using the 'count.seqs' command.

The remaining sequences were aligned against SILVA database (release 119; (*Pruesse et al., 2007*). Sequences were pre-clustered allowing for up to a 2 nt difference between the sequences (*Huse et al., 2010*). Chimeras were removed using UCHIME as implemented in mothur (*Edgar et al., 2011*). Next, sequences were classified with Greengenes database (release gg_13_8_99; bootstrap = 60; *McDonald et al., 2012*), followed by the removal of chloroplast, mitochondria, Archaea, and eukaryote sequences. Further, we found three abundant bacterial families (Dermabacteraceae, Dietziaceae, Brevibacteriaceae) that were present in all disease samples and at high abundance in our negative control. The negative control was a sample containing water as a template for the PCR reaction. As these bacterial families are also known as kit/reagent/lab contaminants, they were excluded from the dataset (*Salter et al., 2014*). Some additional bacterial taxa that were found in high numbers in the negative control with low abundance in coral samples were excluded (Comamonadaceae, Halomonadaceae, Staphylococcaceae). For further analyses, sequences were subsampled to 7,328 sequences per sample, which is the lowest number of sequences in a sample, and then clustered into OTUs (Operational Taxonomic Units) at a 97% similarity cutoff. Reference sequences for each OTU were determined by the most abundant sequence (Data S1). Alpha diversity indices (i.e., Chao1 *Chao, 1984*, Simpson evenness, and Inverse Simpson Index *Simpson, 1949*) were calculated as implemented in mothur.

For detecting similarity of the three main microbial consortium members in BBD microbial mats to previously reported taxa from other studies, the representative sequences of the most abundant OTUs were BLASTed against the NCBI database (https://blast.ncbi.nlm.nih.gov) using a 98% similarity cutoff. Subsequently, our sequences were compared to matches of highly similar bacterial taxa by obtaining the respective coral species, their location, and colony health status. Furthermore, low abundant OTUs not previously reported from BBD, but with related properties to SRB, SOB, or Cyanobacteria were also BLASTed against the NCBI database.

16S rRNA sequences of SOB and SRB were aligned and neighbor-joining trees were constructed based on Jukes-Cantor model with MAFFT (*Katoh, Rozewicki & Yamada, 2017*; *Kuraku et al., 2013*). All positions containing gaps and missing data were excluded and phylogenetic trees were visualized using Archaeopteryx.js.

## RESULTS

### Distribution and prevalence of black band disease on Red Sea reefs

We identified 30 coral genera within transects across 22 reef sites (Fig. 1) with an average density of 16 coral colonies / m$^2$ (SE $\pm$ 1.2) and an average coral cover of 43.8% (SE $\pm$ 4.3). Colonies with BBD were found at 8 of 22 sites (Table 1, Fig. 1). Over all study sites, nine coral genera were infected and include *Astreopora, Coelastrea, Dipsastraea, Gardineroseris, Goniopora, Montipora, Pavona, Platygyra, Psammocora.* Approximately 74,090 colonies were examined for disease and overall BBD prevalence over all sites was low (0.04%) because most sites had no signs of BBD. At the sites where BBD occurred, seven of the eight sites had one to two colonies infected within survey areas (up to 300 m$^2$) (avg. prevalence = 0.064%) and one site had a localized BBD outbreak (Al-Lith fringing reef 1) where 21 infected colonies were found within 49 m$^2$ of the transect (prevalence = 1.7%) and an additional two colonies outside the transect (Table 1). At this site, 18 coral genera were found within transects, but only nine coral genera exhibited signs of disease suggesting differential BBD susceptibility among coral genera ($x^2 = 45.67$ $df = 6$, $P < 0.001$). *Dipsastraea* appeared to be the most susceptible (prevalence = 6.1%) with more diseased colonies than expected based on its abundance within transects (Table 2). *Dipsastraea* represented 18.6% of the coral colonies within transects but 68.2% (15 of 22) of the BBD colonies.

### Bacterial community composition of black band disease microbial mats

Besides the ecological survey of BBD prevalence, we investigated the microbial consortium of the BBD mat of corals from the outbreak site in the southern central Red Sea that was also subject to a bleaching event (Al-Lith fringing reef 1). We assessed whether the same bacterial taxa are associated with BBD in the Red Sea in comparison to other sites globally. Seven coral BBD microbial mat samples from the outbreak site included one colony of *Coelastrea* sp., two colonies of *Dipsastraea* sp., three colonies of *Goniastrea* sp., and one colony of *Platygra* sp., which together yielded 555,093 raw 16S rRNA gene sequences with a mean length of 298 bp (Table 3). After quality filtering and exclusion of chimeras and

**Table 2** Survey of black band disease-affected coral genera at an outbreak site in the southern central Red Sea (Al-Lith fringing reef 1, Saudi Arabia).

| Coral species | No. of coral colonies/survey area (20 m²) | % of coral community | No. of BBD cases/survey area (49 m²) | Prevalence % |
|---|---|---|---|---|
| *Astreopora* | 23 | 4.24 | 1 | 1.8 |
| *Coelastrea*[a] | 0 | – | 1 | – |
| *Dipsastraea* | 101 | 18.63 | 15 | 6.1 |
| *Goniopora*[a] | 0 | – | 1 | – |
| *Montipora* | 11 | 2.03 | 1 | 3.7 |
| *Pavona* | 5 | 0.92 | 1 | 8.2 |
| *Platygyra* | 28 | 5.17 | 1 | 1.5 |
| *Psammocora* | 28 | 5.17 | 2 | 2.9 |
| Other coral genera | 346 | 63.80 | 0 | 0 |
| **Total**[a] | **542** | **100** | **23** | **1.7** |

Notes.

[a] Colonies of *Coelastrea* and *Goniopora* were only found outside the survey area and were not counted towards totals.

**Table 3** Summary of sequencing information and alpha diversity measures of bacterial communities associated with black band disease microbial mats from coral colonies at an outbreak site in the southern central Red Sea (Al-Lith fringing reef 1, Saudi Arabia).

| Sample | No. of sequences | No. of OTUs[a] | Chao1[a] | Inv. Simpson[a] | Simpson evenness[a] |
|---|---|---|---|---|---|
| *Coelastrea* | 17,869 | 149 | 203 | 14.10 | 0.095 |
| *Dipsastraea 1* | 7,328 | 146 | 240 | 22.33 | 0.153 |
| *Dipsastraea 2* | 15,919 | 122 | 160 | 7.71 | 0.063 |
| *Goniastrea 1* | 16,201 | 120 | 152 | 8.18 | 0.068 |
| *Goniastrea 2* | 15,743 | 113 | 161 | 9.49 | 0.084 |
| *Goniastrea 3* | 14,466 | 136 | 159 | 11.57 | 0.085 |
| *Platygra* | 20,037 | 98 | 125 | 8.50 | 0.087 |

Notes.

[a] After subsampling to 7,328 sequences. Total number of OTUs: 315.

contaminant sequences, we retained 107,613 sequences for analysis of the BBD microbial mat microbiome. To assess bacterial community composition, sequences were classified to family level considering bacterial families that comprised >1% of the total sequence reads (Fig. 2). The presence of cyanobacteria, SRB, and SOB was confirmed but at varying abundance. For instance, Cyanobacteria such as Phormidiaceae ranged in proportion between 0 and 3.9%, SRB such as Desulfovibrionaceae between 0.9 and 33.8%, and SOB such as Campylobacteraceae between 15 to 45%. After subsampling to 7,328 sequences per sample, we found 351 distinct OTUs across the entire dataset (Data S1). Species richness (Chao1) and bacterial diversity (Inverse Simpson) were relatively similar between samples, ranging from 98 to 149 OTUs per sample (Table 3).

## Black band disease representative bacterial consortia

We compared the sequences from representative bacterial BBD consortium members found in four coral genera in the southern central Red Sea to sequences obtained from other
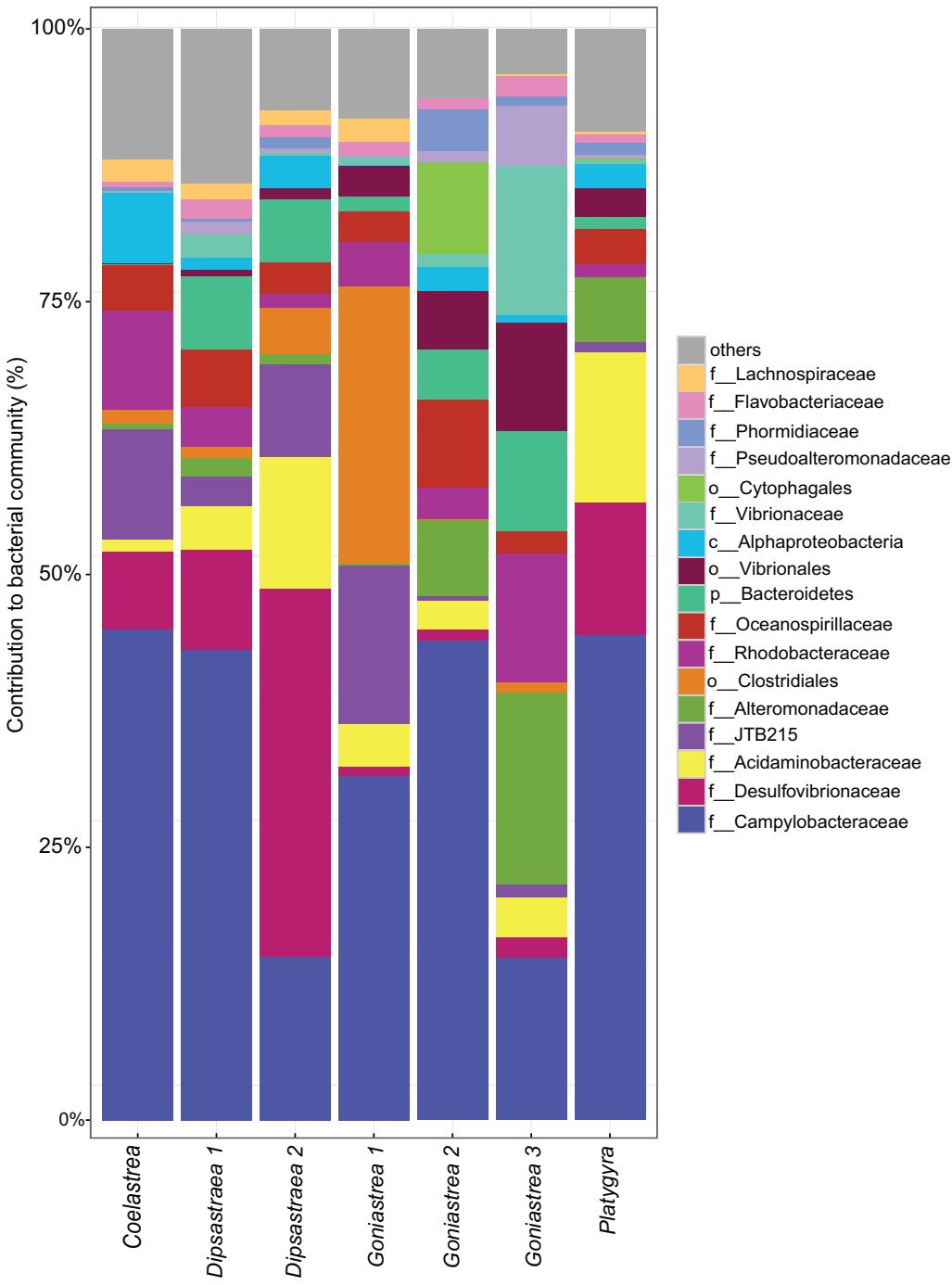

**Figure 2  Bacterial community composition of black band disease microbial mats from four coral genera.** (one colony of *Coelastrea*, two colonies of *Dipsastraea*, three colonies of *Goniastrea*, and one colony of *Platygyra*) from an outbreak site in the southern central Red Sea (Al-Lith fringing reef 1, Saudi Arabia). Taxonomy stacked column plot on the phylogenetic level of family or to lowest resolved taxonomic level (f, family; o, order; p, phylum). Each color represents one of the 17 most abundant families. Remaining taxa are grouped under category 'others'.

locations and coral taxa that were affected by BBD on a global scale. Coral disease microbial mat-associated OTUs that represent the three main bacterial consortium members in BBD were successfully identified in our samples:

### Sulfide oxidizing bacteria (SOB)

*Beggiatoa* sp., a common BBD-SOB member was absent in our samples, despite microscopic white filaments in the disease lesions which suggested its presence. Another SOB-consortium member *Arcobacter* sp. was present in all samples, which has been associated previously with BBD and with white plague disease (WPD)(*Sunagawa et al., 2009*). The SOB-classified OTUs were the most abundant taxa in the dataset. Several OTUs were found to be associated with all coral genera (i.e., *Coelastrea*, *Dipsastraea*, *Goniastrea*, and *Platygra*) with proportions of up to 22.6% in all coral samples (OTU0001, 2, 4, 11, 16: all *Arcobacter* sp., OTU0010: *Sulfurospirillum* sp.). These SOB-associated OTUs were found to be similar to those found in different places around the world (e.g., Philippines (*Garren et al., 2009*) and in the Caribbean including the Netherlands Antilles (*Klaus, Janse & Fouke, 2011*), US Virgin Islands (*Cooney et al., 2002*), and Puerto Rico (*Sunagawa et al., 2009*) (Table 4), where they were associated with varying coral species (Fig. 3A).

### Sulfate-reducing bacteria (SRB)

Two abundant OTUs were found to be associated with BBD samples. These OTUs were annotated to *Desulfovibrio* sp. (OTU0005, OTU0006) with abundance ranges of 0.01–30.7% in all coral samples. Similar SRB-OTUs were found in the Caribbean (*Sekar, Kaczmarsky & Richardson, 2008*; *Sunagawa et al., 2009*) and Japan in different coral species (e.g., *Montipora* sp., *Orbicella faveolata*, and *Siderastrea sidereal*, Table 4). OTU0005 (*Desulfovibrio dechloracetivorans*) clustered together with SRB previously found in corals diseased with WPD (*Sunagawa et al., 2009*) and BBD (*Sekar et al., 2006*), while OTU0006 (*Desulfovibrio marinisediminis*) clustered away (Fig. 3B), indicating that this is not a typical BBD consortium member.

### Cyanobacteria

One cyanobacterium (OTU0023, *Oscillatoria* sp.) was found at proportions of up to 4% in our coral samples. Cyanobacteria of the same genus (99% sequence similarity) have previously been found in BBD infected *Pavona* sp. in the GBR (*Buerger et al., 2016*) and from other regions, e.g., the Caribbean (*Casamatta et al., 2012*), Hawaii (*Aeby et al., 2015b*), and Palau (*Sussman, Bourne & Willis, 2006*) (Table 4).

### Others

Although not belonging to the three main BBD bacterial consortium members, Firmicutes have previously been reported in coral BBD (*Barneah et al., 2007*; *Cooney et al., 2002*; *Klaus, Janse & Fouke, 2011*). Members were also found in our dataset at proportions of up to 24.8% (OTU0003, OTU0009, OTU0013, OTU0018). Furthermore, the Firmicutes-associated OTUs in our data were similar to those found in *Porites* white patch syndrome (PWPS) (*Séré et al., 2013*) and WPD (*Roder et al., 2014*; *Sunagawa et al., 2009*) (Table 4).

We also retrieved sequences of *Vibrio* sp. (OTU0015, OTU0029) from our dataset, at proportions of up to 12.8%. These OTU sequences also had a high similarity (98–99%)
**Table 4** Summary of bacterial taxa (OTUs) associated with black band disease (BBD) in corals from the southern central Red Sea and comparison with similar taxa from around the world, based on BLAST results (accession number, identity) of the BBD consortium of sulfide-oxidizing bacteria (SOB), sulfate-reducing bacteria (SRB), cyanobacteria, Firmicutes, and *Vibrio* sp.

| OTU | Count | Taxonomy | Identity | GenBank Acc No. | Reference | Health state | Host & location |
|-----|-------|----------|----------|-----------------|-----------|--------------|-----------------|
| **SOB** | | | | | | | |
| Otu0001 | 913 | *Arcobacter* sp. | 99% | EF089456 | *Barneah et al. (2007)* | BBD | *Favites* and *Dipsastraea*, Red Sea |
| | | | | KC527436 | *Roder et al. (2014)* | WPD | *Pavona duerdeni* and *Porites lutea*, West Pacific |
| | | | | HM768631 | *Klaus, Janse & Fouke (2011)* | BBD | Faviidae, Meandrinidae, Gorgoniidae, Caribbean |
| Otu0002 | 625 | *Arcobacter* sp. | 99% | GU319311 | *Meron et al. (2010)* | Healthy | *Acropora eurystoma*, Red Sea |
| | | | | FJ203140 | *Sunagawa et al. (2009)* | WPD | *Orbicella faveolata*, Caribbean |
| | | | | AB235414 | *Yasumoto-Hirose et al. (2006)* | – | Non-coral species |
| Otu0004 | 336 | *Arcobacter* sp. | 99% | KT973145 | *Couradeau et al. (2017)* | – | Non-coral species |
| | | | | JF344171 | *Acosta-González, Rosselló-Móra & Marqués (2012)* | – | Non-coral species |
| | | | | FJ949362 | *Suárez-Suárez et al. (2011)* | – | Non-coral species |
| Otu0010 | 246 | *Sulfurospirillum* sp. | 98% | LC026456 | K Yamaki, F Mori, R Ueda, R Kondo, U Umezawa, H Nakata & M Wada (2015, unpublished data) | – | Non-coral species |
| | | | | AF473976 | *Cooney et al. (2002)* | BBD | Faviidae, Caribbean |
| | | | | GU472074 | L Arotsker, D Rasoulouniriana, N Siboni, E Ben-Dov, E Kramarsky-Winter, Y Loya & A Kushmaro (2010, unpublished data) | BBD | – |
| Otu0011 | 208 | *Arcobacter* sp. | 99% | HM768558 | *Klaus, Janse & Fouke (2011)* | BBD | Faviidae, Meandrinidae, and Gorgoniidae, Caribbean |
| | | | | GQ413587 | *Garren et al. (2009)* | – | *Porites cylindrica*, West Pacific |
| Otu0016 | 117 | *Arcobacter* sp. | 98% | LC133150 | S Iehata, Y Mizutani & R Tanaka (2016, unpublished data) | – | Non-coral species |
| | | | | HE804002 | C Chiellini, R Iannelli, F Verni & G Petroni (2012, unpublished data) | – | Non-coral species |
| | | | | KF185679 | J Vojvoda, D Lamy, E Sintes, JA Garcia, V Turk & GJ Herndl (2013, unpublished data) | – | Non-coral species |
| **SRB** | | | | | | | |
| Otu0005 | 322 | *Desulfovibrio dechloracetivorans* | 98% | AB470955 | K Yoshinaga, BE Casareto & Y Suzuki (2008, unpublished data) | Healthy | *Montipora* sp., West Pacific |
| | | | | FJ202627 | *Sunagawa et al. (2009)* | WPD | *Orbicella faveolata*, Caribbean |
| | | | | EF123510 | *Sekar, Kaczmarsky & Richardson (2008)* | BBD | *Siderastrea siderea*, Caribbean |

**Table 4** (*continued*)

| OTU | Count | Taxonomy | Identity | GenBank Acc No. | Reference | Health state | Host & location |
|-----|-------|----------|----------|-----------------|-----------|--------------|-----------------|
| Otu0006 | 294 | *Desulfovibrio marinisediminis* | 99% | MF039931 | R Keren, A Lavy, I Polishchuk, B Pokroy & M Ilan (2017, unpublished data) | – | Non-coral species |
| | | | | KY771114 | H Zouch, F Karray, A Fabrice, S Chifflet, A Hirschler, H Kharrat, W Ben Hania, B Ollivier, S Sayadi & M Quemeneur (2017, unpublished data) | – | Non-coral species |
| | | | | KT373805 | K Alasvand Zarasvand & VR Rai (2015, unpublished data) | – | Non-coral species |
| **Cyanobacteria** | | | | | | | |
| Otu0023 | 81 | *Oscillatoria* sp. | 99% | KU579394 | *Buerger et al. (2016)* | BBD | *Pavona*, Great Barrier Reef |
| | | | | HM768593 | *Klaus, Janse & Fouke (2011)* | BBD | Faviidae, Meandrinidae, Gorgoniidae,Caribbean |
| | | | | GU472422 | L Arotsker, D Rasoulouniriana, N Siboni, E Ben-Dov, E Kramarsky-Winter, Y Loya & A Kushmaro (2010, unpublished data) | BBD | – |
| **Firmicutes** | | | | | | | |
| Otu0003 | 344 | family JTB215 | 99% | DQ647593 | R Guppy & JC Bythell (2006, unpublished data) | – | – |
| | | | | KC527313 | *Roder et al. (2014)* | WPD | *Pavona duerdeni* and *Porites lutea*, Caribbean |
| | | | | HM768569 | *Klaus, Janse & Fouke (2011)* | BBD | Faviidae, Meandrinidae, Gorgoniidae, Caribbean |
| Otu0013 | 199 | *Fusibacter* sp. | 99% | GQ413281 | *Garren et al. (2009)* | – | *Porites cylindrica*, West Pacific |
| | | | | FJ202930 | *Sunagawa et al. (2009)* | WPD | *Orbicella faveolata*, Caribbean |
| Otu0018 | 112 | *Fusibacter* sp. | 99% | KF179748 | *Séré et al. (2013)* | PWPS | *Porites lutea*, Western Indian Ocean |
| | | | | GU472060 | L Arotsker, D Rasoulouniriana, N Siboni, E Ben-Dov, E Kramarsky-Winter, Y Loya & A Kushmaro (2010, unpublished data) | BBD | – |
| | | | | EU780347 | SE Godwin, J Borneman, E Bent & L Pereg-Gerk (2008, unpublished data) | SWS | *Turbinaria mesenterina*, |
| | | | | EF089469 | *Barneah et al. (2007)* | BBD | *Favites, Dipsastraea*, Red Sea |
| Otu0022 | 82 | family Lachnospiraceae | 99% | HM768582 | *Klaus, Janse & Fouke (2011)* | BBD | Faviidae, Meandrinidae, Gorgoniidae, Caribbean |
| | | | 98% | AF473930 | *Cooney et al. (2002)* | BBD | Faviidae, Caribbean |
| | | | | DQ647585 | R Guppy & JC Bythell (2006, unpublished data) | – | – |
| Otu0027 | 62 | *Fusibacter* sp. | 99% | JX391361 | YYK Chan, AL Li, S Gopalakrishnan, RSS Wu, SB Pointing & JMY Chiu (2012, unpublished data) | – | Non-coral species |
| | | | | HM768587 | *Klaus, Janse & Fouke (2011)* | BBD | Faviidae, Meandrinidae, Gorgoniidae, Caribbean |
| | | | | FJ202981 | *Sunagawa et al. (2009)* | WPD | *Orbicella faveolata*, Caribbean |
| Otu0031 | 48 | *WH1-8* sp. | 99% | KF179804 | *Séré et al. (2013)* | PWPS | *Porites lutea*, Western Indian Ocean |
| | | | | KC527300 | *Roder et al. (2014)* | WPD | *Pavona duerdeni* and *Porites lutea*, West Pacific |
| | | | | FJ203165 | *Sunagawa et al. (2009)* | WPD | Faviidae, Caribbean |

Hadaidi et al. (2018), *PeerJ*, DOI 10.7717/peerj.5169
**Table 4** (*continued*)

| OTU | Count | Taxonomy | Identity | GenBank Acc No. | Reference | Health state | Host & location |
|---|---|---|---|---|---|---|---|
| Otu0039 | 30 | *Defluviitalea saccharophila* | 99% | DQ647556 | R Guppy & JC Bythell (2006, unpublished data) | – | – |
| | | | | FJ202907 | *Sunagawa et al. (2009)* | WPD | *Orbicella faveolata*, Caribbean |
| | | | | AF473925 | *Cooney et al. (2002)* | BBD | Faviidae, Caribbean |
| **Vibrio sp.** | | | | | | | |
| Otu0015 | 170 | *Vibrio* sp. | 99% | KT974549 | *Couradeau et al. (2017)* | – | Non-coral species |
| | | | 98% | GU471972 | L Arotsker, D Rasoulouniriana, N Siboni, E Ben-Dov, E Kramarsky-Winter, Y Loya & A Kushmaro (2010, unpublished data) | BBD | – |
| | | | 98% | MF461384 | *Keller-Costa et al. (2017)* | – | *Eunicella labiata* |
| Otu0029 | 56 | order Vibrionales | 99% | JQ727003 | *Witt, Wild & Uthicke (2012)* | – | Non-coral species, GBR |
| | | | | HM768601 | *Klaus, Janse & Fouke (2011)* | BBD | Faviidae, Meandrinidae, Gorgoniidae, Caribbean |
| | | | | FJ202558 | *Sunagawa et al. (2009)* | WPD | *Orbicella faveolata*, Caribbean |

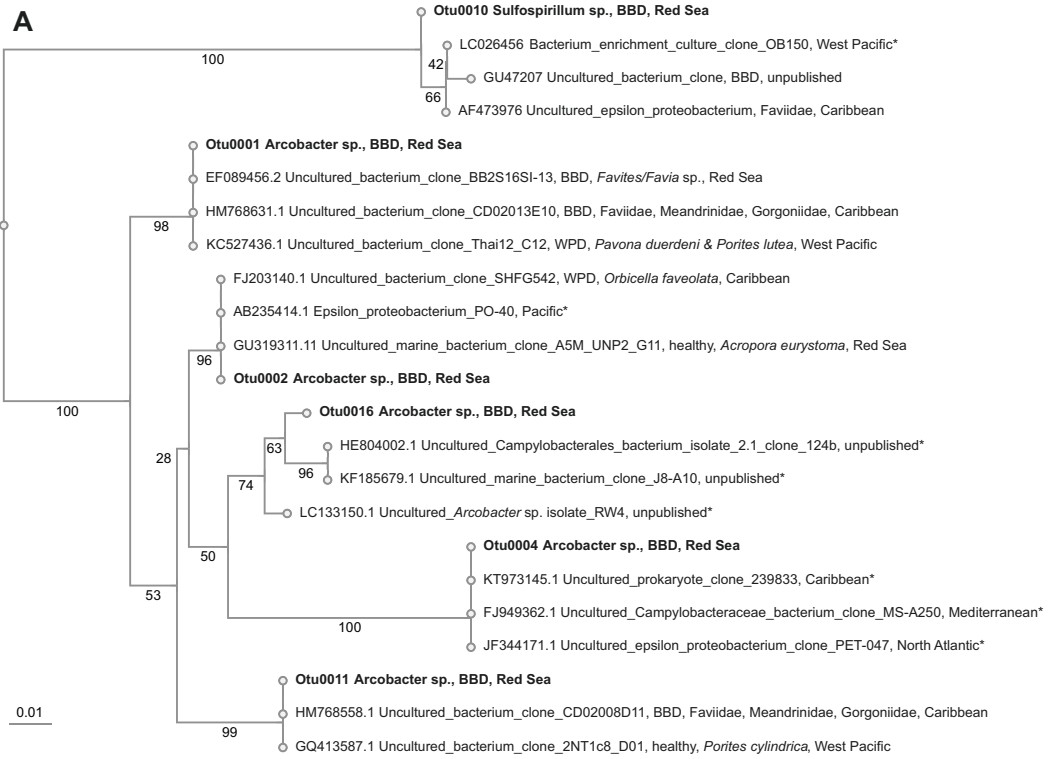

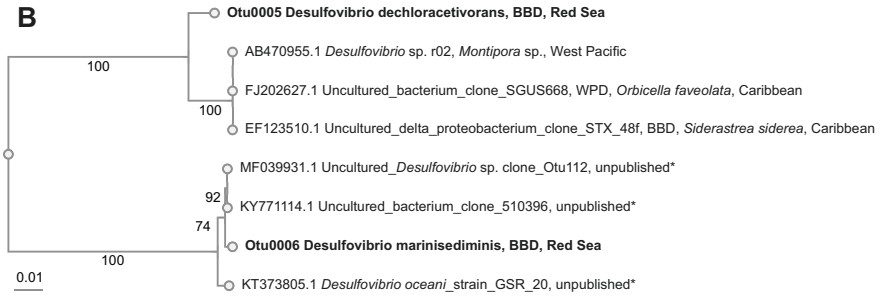

**Figure 3** **Overview and phylogenetic relationship of coral black band disease bacterial consortium members from the southern central Red Sea (Al-Lith, Saudi Arabia) and other regions.** (A) Sulfide-oxidizing bacteria (SOB); (B) Sulfate-reducing bacteria (SRB). Phylogenetic trees were calculated using the neighbor-joining method, bootstrap values are indicated at the branches. The phylogenetic trees show NCBI accession numbers and sample name, health state of the coral species, host name, and region. Sequences from this study are in bold. The '*' indicates that the bacterial species were not found in coral species.

to sequences from BBD and WPD (*Klaus, Janse & Fouke, 2011*; *Sunagawa et al., 2009*) (Table 4).

## DISCUSSION

In this study, we report on the distribution and prevalence of coral black band disease in the Red Sea. Our surveys ranged from 19.9 to 24.1 degrees of latitude and confirm the

presence of BBD across the central Red Sea. Molecular characterization of the bacterial community identified the three main bacterial members of the disease consortium across coral species at a BBD outbreak site in the southern central Red Sea.

## Black band disease distribution and prevalence in the Red Sea in comparison to other global sites

BBD is a global disease found in numerous regions, but its prevalence on coral reefs is generally low compared to other diseases such as white syndrome (WS) (*Dinsdale, 2002*; *Edmunds, 1991*; *Page & Willis, 2006*; *Willis, Page & Dinsdale, 2004*). The low prevalence recorded in this study is similar to levels reported elsewhere across the globe (*Sutherland, Porter & Torres, 2004*) with localized outbreaks of BBD also reported in the GBR (*Sato, Bourne & Willis, 2009*), Hawaii (*Aeby et al., 2015b*), Jamaica (*Bruckner & Bruckner, 1997*), Venezuela (*Rodríguez & Cróquer, 2008*), and the Red Sea (*Al-Moghrabi, 2001*). In the Red Sea, BBD was first discovered in the 1980s (*Antonius, 1981*) and our study confirms that BBD is a chronic threat to coral reefs in the Red Sea with localized outbreaks continuing to occur.

BBD is not a selective disease; multiple species and various levels of severity can affect colonies within and between coral species and across reefs (*Bruckner, Bruckner & Williams, 1997*; *Dinsdale, 2002*; *Green & Bruckner, 2000*; *Peters, 1993*). This was also observed in our study, where multiple species were infected, but with differences in prevalence among coral taxa. At the outbreak site, we found BBD prevalence to be highest in the genus *Dipsastraea*, which suggests that this genus may be an important host for BBD in the Red Sea. Our observations match previous reports and shows that this pattern is consistent through time (*Antonius, 1985*). Interestingly, although differential susceptibility to BBD among coral taxa has been found globally, the most vulnerable taxa differ by region. For example, in the Caribbean *Montastraea/Orbicella* are commonly infected (*Bruckner & Bruckner, 1997*; *Porter et al., 2001*), *Montipora* in Hawaii (*Aeby et al., 2015b*), and *Acropora* on the GBR (*Page & Willis, 2006*). It would be fruitful to examine the underlying defense mechanisms in the different coral taxa that lead to these differences in BBD occurrence.

## BBD, climate change, and coral bleaching

The occurrence of BBD has been linked to elevated seawater temperatures (*Boyett, Bourne & Willis, 2007*; *Kuta & Richardson, 2002*; *Muller & Van Woesik, 2011*). The occurrence of a BBD outbreak during a bleaching event in the present study reflects previous reports from the Caribbean, where the positive correlation between bleaching events and BBD incidence was proposed first (*Brandt & McManus, 2009*; *Cróquer & Weil, 2009*). For instance, in the Florida Reef Tract, the prevalence of BBD increased from 0 to 6.7% following bleaching events in 2014 and 2015 (*Lewis et al., 2017*). Also, *Cróquer & Weil (2009)* found a significant linear correlation between coral bleaching and the prevalence of two other virulent diseases (yellow band disease and white plague) affecting *Montastraea/Orbicella* species. This further supports a strong relationship between bleaching events and the emergence of some coral diseases on a global scale. Understanding how climate change-related thermal anomalies and coral bleaching drive the emergence and virulence of coral diseases is essential for future research.

It has further been suggested that other anthropogenic activities, such as coastal pollution or ocean acidification, contribute to the increase of coral disease incidents (*Jackson et al., 2001*; *Muller et al., 2017*; *Rosenberg & Ben-Haim, 2002*). The surveyed outbreak area was adjacent to the outflow of a large aquaculture facility, which might have further aggravated the effects of the bleaching event due to increased nutrient availability (*Roder et al., 2015*; *Ziegler et al., 2016*). In comparison, other reefs in the Al-Lith area that were farther away from the coast displayed similar levels of bleaching, but BBD prevalence stayed at baseline levels in these locations. This suggests that bleaching alone was not the only factor that could have contributed to the BBD outbreak. The synergistic effects of high temperatures and nutrient pollution find further support in the Caribbean where BBD prevalence increased in reef sites with direct sewage input compared to control sites (*Sekar, Kaczmarsky & Richardson, 2008*) and in the Bahamas where BBD migration was faster in nutrient-enriched areas (*Voss & Richardson, 2006*). Further work is needed to directly examine the relationship between bleaching, nutrient stress, and BBD susceptibility.

## Bacterial community composition of BBD microbial mats from the southern central Red Sea reflects global microbial patterns with local characteristics

Our results verify the presence of the three main consortium members in BBD microbial mats (Cyanobacteria, SOB, SRB) of corals from the southern central Red Sea. We identified *Oscillatoria* sp. as a BBD-associated cyanobacterium, which is similar to the BBD-associated cyanobacteria in other regions of the world (*Aeby et al., 2015b*; *Arotsker et al., 2015*; *Buerger et al., 2016*; *Casamatta et al., 2012*; *Cooney et al., 2002*; *Frias-Lopez et al., 2003*; *Gantar, Sekar & Richardson, 2009*; *Glas et al., 2010*; *Meyer et al., 2016*; *Miller & Richardson, 2011*; *Rasoulouniriana et al., 2009*; *Sato, Willis & Bourne, 2010*; *Sussman, Bourne & Willis, 2006*). However, we retrieved only a low number of cyanobacterial sequences, although cyanobacterial filaments were visually abundant in the sampled microbial mats, which could possibly be related to primer amplification bias. In addition, members of the SOB and SRB functional groups (*Arcobacter* sp. and *Desulfovibrio* sp., respectively) from BBD microbial mats in the southern central Red Sea were similar to those found in other BBD-affected corals worldwide (*Barneah et al., 2007*; *Cooney et al., 2002*; *Klaus, Janse & Fouke, 2011*; *Sekar, Kaczmarsky & Richardson, 2008*). This confirms that BBD-associated bacteria are not restricted to a specific coral species or region (*Barneah et al., 2007*; *Cooney et al., 2002*; *Dinsdale, 2002*; *Frias-Lopez et al., 2003*). Interestingly, we did observe white filaments within lesions that were morphologically similar to *Beggiatoa*, a sulfide-oxidizing bacterium associated with BBD in other regions (*Cooney et al., 2002*; *Miller & Richardson, 2011*; *Sato, Willis & Bourne, 2010*). However, we found no sequences aligning with *Beggiatoa* in our study. This suggests that either the white filaments were not *Beggiatoa* or that the methods used were not adequate to extract and identify *Beggiatoa*. *Aeby et al. (2015b)* sequenced *Beggiatoa* from BBD lesions in Hawaii by first culturing the white filaments from lesions and then using universal bacterial primers 8F and 1513R for sequencing. However, they found that no DNA sequences were available for *Beggiatoa* found in BBD from other

regions even though numerous studies using molecular techniques have been published. Further work is needed to clarify these discrepancies.

Besides the three main bacterial consortium members that dominate BBD microbial mats, we detected other bacterial families as part of the BBD consortium. Members of the Firmicutes were abundant in BBD microbial mats, which is consistent with other studies (*Arotsker et al., 2016*; *Arotsker et al., 2009*; *Barneah et al., 2007*; *Cooney et al., 2002*; *Frias-Lopez et al., 2002*; *Miller & Richardson, 2011*; *Richardson, 2004*; *Sekar, Kaczmarsky & Richardson, 2008*). In addition, we detected the presence of *Vibrio* species. The pathogenicity of this genus has been documented previously in corals and other marine organisms (*Ben-Haim, Zicherman-Keren & Rosenberg, 2003*; *Harvell et al., 1999*; *Kushmaro et al., 1996*), and more broadly *Vibrios* have been characterized as opportunistic taxa (*Cervino et al., 2004*; *Rosenberg & Falkovitz, 2004*; *Thompson et al., 2004*; *Ziegler et al., 2016*). To date it is unknown whether this group plays a role in the etiology of BBD (*Arotsker et al., 2009*; *Barneah et al., 2007*) (*Meyer et al., 2016*), or whether the high number of *Vibrios* could be related to seasonal increases in the coral microbiome and coral bleaching (reviewed in *Rosenberg & Koren, 2006*; *Tout et al., 2015*).

## CONCLUSIONS

Our study represents the first comprehensive assessment of Black Band Disease in the central Red Sea. Elucidation of the bacteria associated with BBD microbial mats of corals at a southern reef site confirms that BBD represents a disease with predictable etiology where the three main bacterial players are globally distributed with regional differences. Notably, our reef survey data, in line with data from other regions, identify BBD as a widespread disease, but as one with low prevalence in comparison to other coral diseases. Additional surveys including other coral diseases as well as pathogen infection experiments with Red Sea corals could further increase our understanding of coral stress tolerance in this understudied coral reef region. Importantly, the prevalence of BBD might increase with ongoing ocean warming and thermal anomalies, as supported by the here-documented disease outbreak coinciding with a thermal anomaly and widespread coral bleaching. The collection of long-term monitoring disease data in the Arabian Seas is important in order to establish baselines, which can then assist in more accurate prediction of disease prevalence and potential impact of climate change on coral communities in this region.

## ACKNOWLEDGEMENTS

We would like to thank the KAUST Coastal and Marine Resources Core Lab (CMOR) for their assistance and support in field operations and the KAUST Bioscience Core Lab (BCL) for sequencing. We wish to thank Craig Michell (KAUST) for sequence library preparation and Nikolaos Zarokanellos (KAUST) for help with Fig. 1.

## Funding

Research reported in this publication was supported by baseline research funds to Christian R. Voolstra and Red Sea Research Center funded project FCC/1/1973-21-01 by KAUST. The funders had no role in study design, data collection and analysis, decision to publish, or preparation of the manuscript.

## Grant Disclosures

The following grant information was disclosed by the authors:
Baseline Research Funds.
KAUST: FCC/1/1973-21-01.

## Competing Interests

The authors declare there are no competing interests.

## Author Contributions

- Ghaida Hadaidi performed the experiments, analyzed the data, prepared figures and/or tables, authored or reviewed drafts of the paper, approved the final draft.
- Maren Ziegler conceived and designed the experiments, performed the experiments, analyzed the data, prepared figures and/or tables, authored or reviewed drafts of the paper, approved the final draft.
- Amanda Shore-Maggio performed the experiments, analyzed the data, approved the final draft.
- Thor Jensen performed the experiments, approved the final draft.
- Greta Aeby and Christian R. Voolstra conceived and designed the experiments, performed the experiments, analyzed the data, contributed reagents/materials/analysis tools, prepared figures and/or tables, authored or reviewed drafts of the paper, approved the final draft.

## DNA Deposition

The following information was supplied regarding the deposition of DNA sequences:
Sequence data determined in this study is available at NCBI under BioProject ID PRJNA436216. Abundant coral bacterial microbiome OTU reference sequences are available under GenBank Accession numbers MH341637–MH341689.

## Data Availability

The OTU abundance table is provided as a Data S1.

## Supplemental Information

Supplemental information for this article can be found online at http://dx.doi.org/10.7717/peerj.5169#supplemental-information.

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
