# Peer review of "Ecological and molecular characterization of a coral black band disease outbreak in the Red Sea during a bleaching event"

_PeerJ, doi:10.7717/peerj.5169_

## Round 0.1 · original submission · Major Revisions

Please go through reviewers' comments carefully and revise your ms accordingly. When you submit your revised ms, please also provide your response to the comments on a point-to-point format.

Reviewer 1 ·

Basic reporting

No Comment

Experimental design

I can individuate two separate parts in this manuscript. The part related to microbial analyses seems solid with correct methods and analyses and clear results. The problem I see is that the question on this part is not well stated. It looks like the objective of the paper is quantifying the occurrence and then they also did a characterisation of the microbial fauna. I suggest to well state the objectives of the microbial fauna characterisation because it would give more value to the paper, in fact the comparison with microbial fauna associated to other BDD is interesting and worth to valorise.
The part related to community composition and incidence of the BBD is less solid . There are some problems in the experimental design. For example it is known that BDD is light, temperature and consequently depth dependent. Authors says that their transects vary between 6 and 25 meters (lines 113-114), which is a huge variation. Data on BDD occurrence should be associated with depth, at least, in order to understand if the variability encountered is due to the depth or to the characteristics of the site.
Regarding the results and discussion I have some doubts on the disease %: authors found a 0.04% infection, commenting that those results are similar to other regions, but I see that values in different paper are way higher. Johan et al 2016 (Diversity) and 2015 (Hayati Journal of Biosciences) report variable infection % with high differences between reef crest (12%) and reef slope (0.05%). Results of the present work seems more comparable with reef slope data than with reef crest ones, so definitively more precise data on the depth of the transects are needed. Also other authors (e.g. Sato et al 2009-, proceedings royal society B) report higher percentages. Looking at the literature, looks like the data of the present paper show a lower infection percentage than other sites.

Validity of the findings

Conclusions on the most affected taxa should take into account the frequency of the infected taxa. If a taxa is more present it have more probability to be infected, also if the infection is not selective. Daya on the frequency of the taxa should be compared to the frequency of infected colonies of that taxa in order to normalise the infection rate to the abundance of the taxa.

·

Basic reporting

Hadaidi et al. conducted a series of coral disease surveys along the coast of the Red Sea, where there are no published baseline levels of black band disease. In addition, the authors collected samples of the black band disease mat and characterized the bacterial community to compare with other samples reported around the world. The manuscript is very well written in clear and concise English and very little grammatical editing is needed.
The introduction covers a decent amount of background information setting the stage for the study. The references cited are relevant and well placed. There are a couple of references that could be included such as (Muller and van Woesik 2010 in JMBE and Muller et al. 2017 in PLoSONE). You may already be aware of these and left them out for relevance reasons, and that is fine, but they could be explored for inclusion relative to the environmental influence on BBD disease (2010 paper) and the microbial community (2017 paper).

The tables and figures are appropriate and well made, but need some adjustments (see general comments).

There are no explicit hypotheses stated, but there are identified objectives and the results are relevant to them.

Experimental design

When did these surveys actually take place? I can’t seem to find the dates in the text.
The research objective is clear and fulfills an identified research gap, providing baseline data for BBD within the Red Sea and helps to characterize the consortium of bacteria that creates BBD within this region. There are certain limitations to the experimental design, however. Site selection seems haphazard at best and there is no depth stratification within the sampling design. Since there are survey replicates within each of the three regions, perhaps regional comparisons of the coral disease prevalence could be conducted?
The lack of replication of the BBD samples within species is an additional concern as well as the collection of samples from only one site. The authors obviously take care not to run stats or extrapolate information beyond what is possible within the limited samples, which I appreciate, but this also limits the amount of impactful science that can come from this publication.
Since the sites and transects varied by depth, it would be interesting to also explore the relationship between prevalence and depth. Also, how about regional comparisons of disease prevalence?

Validity of the findings

You mention this outbreak site that occurred during a bleaching event, but there is no bleaching severity information. Other sites bleached and had less levels of BBD or BBD absent, so how would bleaching be related to this particular site’s prevalence? Can you get bleaching severity data retroactively to explore whether this has any association with black band disease severity?
I am also wondering about the sewage outflow information. Any chance you could quantify that through nutrient data? Otherwise, I would simply discuss the fact that a bleaching event was occurring and this happened to be located near a sewage outflow in the discussion as additional information on the site characteristics where the outbreak occurred. These two factors are purely speculation in their relation to disease severity (since they were not directly measured) and this should be explicit within the discussion.
Summary of thoughts:
1. That data provides an identified data gap within the Red Sea
2. The sampling design limits the impact of the science produced from this study
3. Tables and figures need further clarification and additional information
4. Speculation that bleaching may be associated with the outbreak site does not have data to support these results directly. Just needs to be placed in context better.

Additional comments

Line 68: change ‘may results in’ to ‘may result in’
Lines 107 to 109: report the actual permit numbers; identify what the acronym KAUST stands for.
Line 113: if area covered substantially differed among sites then this should be provided as supplemental info
Line 126: the sample collection calls these samples ‘disease lesions’. To me, the lesion would be the area of denuded skeleton caused by the BBD consortium, whereas the actual sampled BBD would simply be the cyanobacterial dominated mat that creates the lesion area. I recommend changing this for clarity.
Line 167: Can you provide more information on the negative control, what was actually used as the negative ‘sample’

Table 1. The ‘GPS’ label should be units; Lat, Long or decimal degrees, or UTMs…whatever these actually represent.
Table 2. I am a bit confused about the coral numbers used in the prevalence calculation. The total number of colonies was surveyed in a smaller area than the disease survey (25 x 1 m vs. 25 x 6 m) and the total number of corals counted within the coral surveyed area should be multiplied by 6 in order to get an estimated total number of corals, correct? However, the prevalence values calculated within this table seem to reflect the original number of corals within the 25 x 1 meter transect. Also, within this table the area of the surveys seems off. The coral survey area is reported as 20 m2, but should be 25 m2 and the disease survey area is reported as 49 m2, but should be 150 m2 (25 x 6).
Table 3. States there are sequencing statistics, but really you are just presenting qualitative information.
Table 4. OTU need to be defined as operational taxonomic unit. This is also where I am surprised that sequences from Muller et al. 2017 are not presented (i.e., did not pop up within your searches), and maybe this is an issue with accessibility within NCBI. Also, how did you pick which accession numbers to present? Was it top three best match to your sequence?
Figure 1. Where is the Jeddah site listed in Table 1?
Figure 2. How did you define a dominant taxonomic group? This should be described in the text of the paper as well. Also, what is the family identified as JTB215?
Figure 3. Could you identify sequences that come from the present study?

---

## Round 0.2 · accepted · Accept

I am happy to accept your ms at its present form.

# ·

Basic reporting

No comment

Experimental design

No comment

Validity of the findings

No comment

Additional comments

The authors have addressed all of my edits and suggestions, included clarifying text, added information for ease of interpretation, and qualified some conclusions. I have no other suggestions for the authors and recommend publication at this time.